# Urinary Inflammatory and Oxidative Stress Biomarkers as Indicators for the Clinical Management of Benign Prostatic Hyperplasia

**DOI:** 10.3390/ijms26136516

**Published:** 2025-07-06

**Authors:** Yuan-Hong Jiang, Jimmy Lee, Hann-Chorng Kuo, Ya-Hui Wu

**Affiliations:** 1Department of Urology, Hualien Tzu Chi Hospital, Buddhist Tzu Chi Medical Foundation, Hualien 97004, Taiwan; hck@tzuchi.com.tw (H.-C.K.); ryoma499@gmail.com (Y.-H.W.); 2Department of Urology, School of Medicine, Tzu Chi University, Hualien 97004, Taiwan; 3Graduate Institute of Pathology, National Taiwan University College of Medicine, Taipei 10051, Taiwan; jimmy720110@ntuh.gov.tw; 4Pathology Department, National Taiwan University Hospital, Taipei 10051, Taiwan

**Keywords:** BPH, urinary biomarker, oxidative stress, inflammatory cytokine

## Abstract

Oxidative stress and hypoxia-induced inflammation contribute to benign prostatic hyperplasia (BPH) progression. This study investigated the roles of urinary inflammatory and oxidative stress biomarkers in BPH patients. This prospective study enrolled 62 clinical BPH patients (33 treated medically, 29 surgically) and 20 controls. Symptom scores, uroflowmetry, and urinary biomarker levels were assessed at baseline and three months post-treatment. Before treatment, BPH patients exhibited elevated urinary levels of total antioxidant capacity (TAC), PGE2, IL-1β, and IL-6. Post-treatment, successful outcomes were reported in 63.6% of the medical treatment group and 86.2% of the surgical treatment group, with improvements in symptom scores and urinary flow rate, along with reductions in urinary 8-isoprostane, TAC, and IL-1β. Prior to treatment, voiding efficiency (VE) was negatively correlated with urinary IL-1β, IL-6, and IL-8 levels, while bladder wall thickness was positively correlated with TAC. After treatment, changes in VE were negatively correlated with changes in IL-1β, and changes in post-void residual urine were positively correlated with changes in IL-1β, IL-6, IL-8, and TNF-α. Urinary inflammatory and oxidative stress biomarkers may serve as non-invasive indicators of disease severity and treatment response in clinical BPH. Their significant correlations with clinical improvements underscore their potential utility in monitoring treatment efficacy.

## 1. Introduction

Lower urinary tract symptoms (LUTS) are highly prevalent and affect more than 60% of men over 40 years of age [1,2]. Clinical benign prostatic hyperplasia (BPH), defined as prostate adenoma/hyperplasia causing bladder outlet obstruction (BOO) [2], is a common cause of male LUTS [3]. BOO can lead to progressive bladder tissue remodeling and potentially serious impairments of the upper urinary tract [4,5]. The development and progression of BOO and its associated urinary dysfunctions are profoundly influenced by cyclic ischemia–reperfusion injury, in which elevated intravesical pressure during voiding induces bladder wall ischemia, followed by reperfusion that generates oxidative stress and triggers hypoxia-related inflammation. This inflammatory microenvironment promotes fibrosis and structural remodeling of bladder tissue, contributing to long-term dysfunction. [5,6].

Oxidative stress, characterized by an imbalance between reactive oxygen species (ROS) and antioxidant defenses, and chronic inflammation have both been increasingly recognized as key contributors in the pathogenesis of BPH [7,8,9]. Prostatic inflammation is driven by multiple overlapping mechanisms, including infection, autoimmune reactions, hormonal alterations, pelvic ischemia, and urine reflux into the prostatic ducts [8]. This inflammatory process contributes to BPH severity by promoting prostatic enlargement and BOO, and inducing storage symptoms. Both chronic inflammation and oxidative stress are thought to actively contribute to BPH progression. Furthermore, tissue hypoxia resulting from BOO is believed to be a major driver of disease advancement, particularly in the context of bladder wall remodeling. Emerging evidence highlights the need for future research to validate oxidative stress-related biomarkers targeting these underlying pathophysiological processes in BPH [9,10].

The significance of oxidative stress biomarkers in BOO, such as 8-hydroxy-2-deoxyguanosine (8-OHdG), F2-isoprostane, and total antioxidant capacity (TAC), was comprehensively reviewed. The alterations in oxidative stress biomarkers associated with BOO and the correlation between oxidative stress and BOO-related urinary dysfunctions were elucidated [6]. Pro-inflammatory cytokines, including tumor necrosis factor-α (TNF-α), IL-1β, IL-6, and IL-8, are crucial in hypoxia-related inflammation [11,12], potentially playing a significant role in the progression of BOO. Recently, urinary oxidative stress and inflammatory biomarker profiles were found to be different among male LUTS with different etiologies, suggesting the potential diagnostic utility of these urinary biomarkers [13].

A three-stage model of bladder wall remodeling due to BOO has been hypothesized, which includes an initial hypertrophy phase, followed by a compensation phase, and ultimately progressing to a decompensation phase [5]. In this model, tissue hypoxia is pivotal in driving disease progression. Early medical or surgical intervention could potentially halt this progression. Biomarkers indicative of oxidative stress may offer valuable insights for the diagnosis and intervention approaches, and become a focus of growing research interest.

In this study, we aimed to investigate the roles of urinary inflammatory and oxidative stress biomarkers in clinical BPH patients, examining their correlations with clinical characteristics and treatment outcomes.

## 2. Results

### 2.1. Baseline Clinical Characteristics and Urinary Biomarker Profiles

The study enrolled 62 clinical BPH patients (mean age 65.33 ± 6.84 years) and 20 non-age-matched controls (mean age 38.0 ± 7.9 years). Compared with controls, BPH patients exhibited significantly thicker bladder walls (bladder wall thickness [BWT]: 2.48 ± 0.72 mm vs. 1.58 ± 0.33 mm; detrusor wall thickness [DWT]: 1.13 ± 0.48 mm vs. 0.69 ± 0.20 mm), higher International Prostate Symptom Scores (IPSS: 17.42 ± 7.11 vs. 1.20 ± 1.20), lower maximal urinary flow rates (Qmax: 9.07 ± 3.79 mL/s vs. 24.72 ± 7.33 mL/s), lower corrected Qmax (cQmax: 0.57 ± 0.24 vs. 1.30 ± 0.72), smaller voided volumes (Vol: 241.89 ± 120.78 mL vs. 425.1 ± 153.52 mL), and reduced voiding efficiency (VE: 0.87 ± 0.13 vs. 0.95 ± 0.08). Prostate volume was measured only in BPH patients, with a mean of 42.76 ± 14.5 mL (Table 1). No significant intergroup differences were observed between the medical treatment and surgical treatment cohorts in clinical characteristics, symptom scores, or uroflowmetry parameters.

At baseline, urinary biomarker analysis showed that clinical BPH patients had significantly higher levels of TAC, prostaglandin E2 (PGE2), IL-1β, and IL-6 compared to controls (Table 2). Post-hoc power analysis reported 8.7, 18, 65.1, and 48.8% power (with an alpha value of 0.05) in the evaluation of TAC, PGE2, IL-1β, and IL-6 levels, respectively. Among clinical BPH patients, the surgical treatment group exhibited significantly higher baseline urinary PGE2 levels and lower IL-1β levels than the medical treatment group.

### 2.2. Treatment Outcomes and Changes in Urinary Biomarkers

Following three-month treatment, successful outcomes were achieved in 63.6% of the medical treatment group and 86.2% of the surgical treatment group, with both groups exhibiting significant improvements in symptom scores and uroflowmetry measures (both Qmax and cQmax).

Post-treatment, significant reductions in urinary 8-isoprostane, TAC, and IL-1β levels were observed across the cohort. In subgroup analyses, the medical treatment group showed significant decreases in urinary IL-1β, IL-8, and TNF-α levels, whereas the surgical treatment group demonstrated significant declines in TAC and PGE2 (Table 2).

Within the medical treatment group, 63.6% (21 of 33) of patients achieved successful outcomes (Table 3). These patients experienced substantial improvements in symptom scores and marked reductions in urinary 8-isoprostane, PGE2, IL-1β, and IL-8 levels. Conversely, patients with unsuccessful outcomes exhibited no significant changes in symptom scores, uroflowmetry parameters, or urinary biomarker profiles, except for a decrease in TNF-α levels.

### 2.3. Correlation Analyses Between Clinical Characteristics and Urinary Biomarkers

Correlation analysis revealed significant associations between clinical parameters and urinary biomarker levels at baseline in the overall clinical BPH cohort (Figure 1). Specifically, VE was negatively correlated with IL-1β, IL-6, and IL-8 levels, while BWT was positively correlated with TAC. After treatment, changes in VE were negatively correlated with changes in IL-1β, and changes in post-void residual urine (PVR) were positively correlated with changes in IL-1β, IL-6, IL-8, and TNF-α (Figure 2). Given the non-normal and right-skewed distribution of IL-1β, IL-6, and IL-8 observed in Figure 1A–C and Figure 2A–B, additional analyses using Spearman’s rank correlation were performed. These supported the directionality of the original findings, although statistical significance was not reached, likely reflecting weaker associations under non-parametric analysis.

Within the medical treatment group, baseline correlation analysis showed that VE was negatively correlated with IL-1β, IL-6, IL-8, and TNF-α levels, whereas BWT was positively correlated with TAC (Appendix A). Following treatment, changes in IPSS voiding subscore (IPSS-V) were positively correlated with changes in IL-1β, IL-6, IL-8, and TNF-α levels; changes in VE were negatively correlated with these biomarkers. Moreover, changes in cQmax were negatively correlated with changes in IL-1β and IL-8 levels (Appendix A).

Appendix A presents baseline and post-treatment correlation analyses between clinical characteristics and urinary biomarker levels in the surgical treatment group.

## 3. Discussion

This study demonstrated significant correlations between clinical characteristics and urinary biomarker levels, as well as between their respective changes following treatment, in clinically diagnosed BPH patients undergoing medical or surgical interventions. Notably, a marked decrease in urinary biomarkers was observed in BPH patients after treatment. In particular, within the medical treatment group, a greater number of urinary biomarkers showed significant reductions in patients who achieved successful outcomes compared to those with unsuccessful outcomes. These findings suggest that urinary biomarkers may serve as potential indicators of disease severity and treatment response. Furthermore, the post-treatment reductions in urinary biomarkers were aligned with improvements in clinical symptom scores and enhanced urinary flow rates, further supporting the notion that oxidative stress and inflammation play critical roles in the pathophysiology of clinical BPH [8,9].

The complex interplay between chronic inflammation and oxidative stress contributes to the progression of BPH [9]. The detection of urinary inflammatory mediators in BPH patients has the potential to distinguish BPH-related pathologies, assess the risk of disease progression, and enable personalized management of BPH-associated LUTS [10,14]. In this study, prior to treatment, clinical BPH patients exhibited significantly higher urinary levels of TAC, PGE2, IL-1β, and IL-6 compared to controls, supporting the concept of an elevated oxidative and inflammatory state in this patient population. Furthermore, VE negatively correlated with hypoxia-related inflammatory biomarkers (IL-1β, IL-6, and IL-8) (Figure 1A–C), suggesting that increased hypoxia-related inflammation is associated with impaired bladder function. This is in line with previous findings that urinary levels of chemokines were associated with varying degrees of LUTS severity [15]. Additionally, a positive correlation between BWT and urinary TAC levels was observed (Figure 1D), suggestive of a structural adaptation of the bladder wall in response to oxidative stress. These findings are consistent with the proposed sequence whereby BOO leads to tissue hypoxia and oxidative stress, which in turn drive the structural remodeling of the bladder wall, manifesting as increased BWT. This remodeling includes smooth muscle hypertrophy, collagen accumulation, and mitochondrial dysfunction, as previously described [5,16].

In this study, after treatment for BPH, changes in VE were negatively correlated with changes in IL-1β, and changes in PVR were positively correlated with changes in IL-1β, IL-6, IL-8, and TNF-α (Figure 2). These findings suggest that improvements in bladder function are associated with reductions in urinary inflammatory biomarkers. Based on these observations, we propose that the treatment of BPH alleviates BOO, thereby improving bladder emptying efficiency and ameliorating oxidative stress and hypoxia-induced inflammation within the urinary bladder, as reflected by decreases in urinary biomarker levels. This interpretation is supported by recent evidence showing that surgical de-obstruction in BPH patients leads to a substantial, although incomplete, reversal of bladder remodeling at the molecular level, alongside improvements in urodynamic parameters [17]. Animal studies further support the reversibility of BOO-induced inflammation and oxidative stress. The relief of obstruction has been shown to reduce inflammatory cytokines and regulatory T cells [18], while oxidative stress markers such as 8-OHdG and malondialdehyde progressively return toward baseline following de-obstruction [19,20].

Notably, differences in urinary biomarker changes were observed between the medical treatment and surgical treatment groups in this study. Medical treatment was associated with significant reductions in inflammatory markers (IL-1β, IL-8, and TNF-α), suggesting that alleviating inflammation plays a major role in clinical improvement. This interpretation is supported by both clinical and experimental studies. Phytotherapeutic agents such as Eviprostat have been shown to reduce urinary oxidative stress markers and improve LUTS [21]. In addition, α1-adrenoreceptor antagonists like silodosin attenuate urine-induced prostatic inflammation and oxidative stress by improving prostatic microcirculation [22].

Conversely, in this study, surgical treatment led to greater decreases in oxidative stress markers (TAC) and PGE2, reflecting the more significant mechanical relief of BOO and restoration of bladder oxygenation. This is in line with recent evidence showing that surgical de-obstruction of the bladder outlet leads to the reversal of bladder wall remodeling, as demonstrated by normalized transcriptional regulators and improved urodynamic parameters [17]. Similar reductions in urinary nerve growth factor and improvements in LUTS after TURP have also been reported [23]. These distinct biomarker response patterns, the inflammation-focused effect of medical treatment and the oxidative stress-related improvements seen after surgery, highlight the differing biological responses associated with medical and surgical treatments for BPH.

This study supports the hypothesis of the impact of BOO on bladder wall remodeling, delineated into three distinct phases: hypertrophy, compensation, and decompensation [5]. Tissue hypoxia, driven by cyclic ischemia–reperfusion injury, plays a critical role in this progression by activating inflammatory pathways and influencing signaling pathways related to angiogenesis, cell proliferation, and extracellular matrix remodeling. These molecular and physiological alterations result in morphological changes and functional impairments in bladder voiding, which may be reflected by changes in urinary biomarker profiles. Our findings align with this pathophysiological model, as elevated urinary levels of inflammatory markers in BPH patients were associated with impaired voiding function, such as lower VE. Notably, distinct urinary biomarker response patterns were observed between medical treatment and surgical treatment groups, suggesting different mechanisms of therapeutic effect—primarily inflammation resolution following medical therapy and oxidative stress reduction after surgical de-obstruction.

This study had several limitations. First, the follow-up period was relatively short, and the sample size was limited, particularly after stratifying patients into medical and surgical groups, which may have affected the statistical power to detect subgroup differences. Second, there might be inter-individual and intra-individual variations that were not fully controlled. Third, our controls were stringently selected based on symptom scores and uroflowmetry to best approximate normal lower urinary tract function. As a result, age-matching was not performed, and the age difference between groups may introduce bias. Fourth, follow-up urinary biomarker measurements were not performed in the control group, as these participants were asymptomatic and not undergoing any clinical intervention. This design aimed to minimize unnecessary procedures and reduce participant burden, but limits the interpretation of intra-individual biomarker variability over time in controls. Finally, treatment selection was based on patient preference, which could introduce selection bias. In addition, future studies should consider incorporating additional assays—such as the direct quantification of ROS or the use of cumulative ROS detection kits—to better delineate oxidative stress mechanisms.

## 4. Materials and Methods

### 4.1. Patients

From November 2020 to July 2022, we prospectively enrolled 62 male patients with voiding-symptom-predominant LUTS and the clinical diagnosis of BPH at the department of urology of a single medical center. The inclusion criteria included age ≥ 40 years of age, IPSS ≥ 12 points, TPV > 30 mL, IPSS-V > IPSS storage subscore (IPSS-S), and Qmax < 15 mL/s with a bladder capacity (defined as Vol plus PVR) ≥ 150 mL. Exclusion criteria included active urinary tract infection, acute or chronic prostatitis, interstitial cystitis, neurogenic voiding dysfunction, urinary tract urolithiasis, a history of urinary tract malignancy or tuberculosis, a history of urinary tract surgery/or traumatic injury, a history of urethral stricture, a history of nephrotic or nephritic syndrome, and impaired renal function (serum Cre > 2.0 mg/dL).

We invited 20 healthy men, who were ready to receive circumcision, vasectomy, or inguinal hernia repair in the same department of urology, to serve as controls. Eligible controls were aged ≥ 20 years old, without significant storage or voiding symptoms (IPSS < 6 points), and with normal uroflowmetry results (defined as Qmax ≥ 18 mL/s, Vol ≥ 350mL, and PVR < 50 mL). A detailed flowchart of the study enrollment process is presented in Appendix A.

### 4.2. Clinical Assessment and Follow-Up

All study patients and controls received clinical assessment on enrollment, including IPSS, IPSS-V, IPSS-S, quality of life score, overactive bladder symptoms score, the measurement of DWT and BWT [24], and uroflowmetry and PVR. Study patients but not controls received a trans-rectal ultrasound of the prostate for the measurement of the prostate size.

Study patients were divided into medical treatment and surgical treatment groups. Medical treatment group patients were treatment-naïve and received alpha blockers (including options such as tamsulosin and silodosin) for 3 months; 5-alpha reductase inhibitors were not administered in this study. Surgical treatment group patients had refractory LUTS and were ready to receive the surgery of transurethral resection of the prostate. Clinical symptom scores, uroflowmetry and PVR, and global response assessment (GRA) for treatment outcome evaluation were assessed at 3 months after treatment. GRA was categorized as −3, −2, −1, 0, 1, 2, and 3, which indicated markedly worse to markedly improved status based on satisfaction after treatment. A successful outcome was defined as a GRA score of ≥ 2 (moderately and markedly improved) [25].

### 4.3. Assessment of Urinary Biomarker Levels

Urine samples were collected from study patients and controls on enrollment, and from study patients at 3 months after treatment. Urine was self-voided by patients who had a full bladder sensation. Then, urinalysis was performed simultaneously to confirm an infection-free status before urine samples were stored. The preparation procedures for urine samples were similar to those reported in the previous study [26]. In total, 50 mL urine samples were placed on ice immediately and transferred to the laboratory for preparation. The samples were centrifuged at 1800 rpm for 10 min at 4 °C. The supernatant was separated into aliquots in 1.5 mL tubes (1 mL per tube) and was preserved in a freezer at −80 °C. Before further analyses were performed, the frozen urine samples were centrifuged at 12,000 rpm for 15 min at 4 °C, and the supernatants were used for subsequent evaluations.

### 4.4. Quantification of Urinary Oxidative Stress Biomarkers

The targeted analytes of oxidative stress biomarkers included 8-OHdG, 8-isoprostane, and TAC. The quantification of 8-OHdG, 8-isoprostane, and TAC in urine samples was performed in accordance with the respective manufacturer’s instructions (8-OHdG ELISA Kit, BioVision, Waltham, MA, USA; 8 isoprostane ELIZA kit, Enzo Life Sciences, Farmingdale, NY, USA; Total Antioxidant Capacity Assay Kit, abcam, Cambridge, UK). The laboratory procedures for the quantification of these targeted analytes were similar to those reported in the previous study [26].

### 4.5. Quantification of Urinary Inflammatory Biomarkers

The targeted analytes of hypoxia-related inflammatory biomarkers included IL-1β, IL-6, IL-8, and TNF-α, which were assayed using a Milliplex^®^ (Darmstadt, Germany) human cytokine/chemokine magnetic bead-based panel kit (catalog number HCYTMAG-60K, Millipore, Darmstadt, Germany). The following laboratory procedures for the quantification of these targeted analytes were similar to those reported in previous studies [26,27].

Urinary PGE2 level was measured using a high-sensitivity ELISA kit (Cayman, MI, USA), according to the manufacturer’s instructions. The laboratory procedures for the quantification of PGE2 were similar to those reported in the previous study [28].

### 4.6. Statistical Analysis

Continuous variables in clinical assessment were presented as means ± standard deviations and categorical variables as numbers and percentages. The data for urinary biomarker levels were presented with the median and interquartile range. Continuous clinical data were compared using independent *t*-tests for two-group comparisons (e.g., study vs. control, and medical treatment vs. surgical treatment). The levels of urinary biomarkers for comparison were analyzed using the Mann–Whitney U test for two-group comparisons. Prior to correlation analysis, the normality of variable distributions was assessed. Pearson correlation analysis was applied for normally distributed variables, while Spearman’s rank correlation was used for variables with non-normal, right-skewed distributions (e.g., IL-1β, IL-6, and IL-8). A post-hoc power calculation was performed for biomarkers with significant differences between the study and control groups. All calculations were performed using SPSS Statistics software for Windows version 20.0 (IBM Corp., Armonk, NY, USA). If the *p*-value is less than 0.05, the difference is considered statistically significant.

## 5. Conclusions

This study provides evidence of the diagnostic and prognostic values of urinary inflammatory and oxidative stress biomarkers in managing clinical BPH patients. These biomarkers may serve as indicators of disease severity and treatment response, offering insights into the underlying pathophysiological mechanisms and the effectiveness of both medical and surgical interventions.

## Figures and Tables

**Figure 1 ijms-26-06516-f001:**
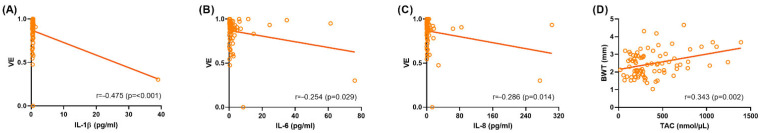
Scatter plots showing the correlations of the clinical characteristics and the baseline urinary biomarker levels in the clinical BPH (both medical and surgical groups) patients. Correlations between VE and IL-1β (**A**), VE and IL-6 (**B**), VE and IL-8 (**C**), and BWT and TAC (**D**). VE, voiding efficiency; BWT, bladder wall thickness; and TAC, total antioxidant capacity.

**Figure 2 ijms-26-06516-f002:**
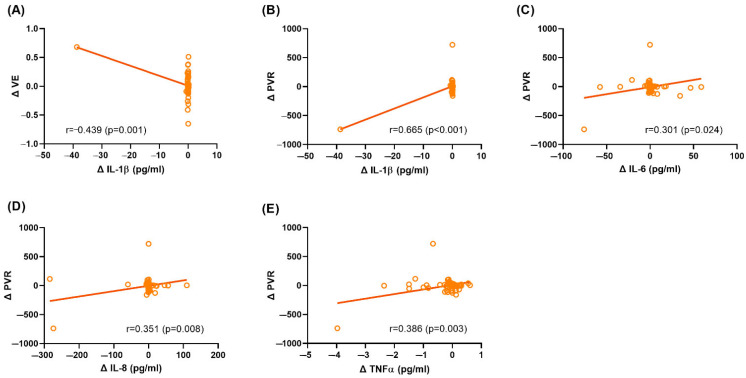
Scatter plots showing the correlations of the changes in clinical characteristics and the changes in urine biomarker levels in clinical BPH (both medical and surgical groups) patients. Correlations between ΔVE and ΔIL-1β (**A**), ΔPVR and ΔIL-1β (**B**), ΔPVR and ΔIL-6 (**C**), ΔPVR and ΔIL-8 (**D**), and ΔPVR and ΔTNF-α (**E**). Δ, change in [specific variable]; VE, voiding efficiency; PVR, post-void residual urine; and TNF-α, tumor necrosis factor-α.

**Table 1 ijms-26-06516-t001:** Clinical characteristics, symptom scores, and uroflowmetry data between clinical BPH patients and controls.

			Clinical BPH				
		Medical Treatment Group N = 33	Surgical Treatment Group N = 29	Overall N = 62	Control N = 20	*p*-Value $	*p*-Value #
Age		64.28 ± 7.12	66.88 ± 6.19	65.33 ± 6.84	38.0 ± 7.9	<0.001	0.087
PSA (ng/mL)		2.41 ± 1.95	2.95 ± 2.66	2.63 ± 2.26		0.287	0.287
TPV (mL)		41.4 ± 13.05	44.76 ± 16.41	42.76 ± 14.5		0.300	0.300
TZI		0.41 ± 0.11	0.44 ± 0.11	0.43 ± 0.11		0.230	0.230
DWT (mm)		1.17 ± 0.54	1.08 ± 0.38	1.13 ± 0.48	0.69 ± 0.20	<0.001	0.410
BWT (mm)		2.53 ± 0.81	2.42 ± 0.57	2.48 ± 0.72	1.58 ± 0.33	<0.001	0.467
Symptom scores							
IPSS-V	Baseline	11.08 ± 5.31	12.35 ± 4.91	11.60 ± 5.16	0.50 ± 0.89	<0.001	0.270
	Δ at 3mo	−3.48 ± 5.96 *	−9.62 ± 6.55 *	−6.35 ± 6.92 *			<0.001
IPSS-S	Baseline	5.20 ± 3.84	6.74 ± 4.19	5.82 ± 4.04	0.70 ± 1.030	<0.001	0.087
	Δ at 3mo	−1.91 ± 3.6 *	−3.17 ± 4.18 *	−2.50 ± 3.90 *			0.206
IPSS	Baseline	16.28 ± 7	19.09 ± 7.03	17.42 ± 7.11	1.20 ± 1.20	<0.001	0.075
	Δ at 3mo	−5.39 ± 7.16 *	−12.79 ± 9.50 *	−8.85 ± 9.06 *			0.001
QoL	Baseline	3.62 ± 1.1	3.97 ± 1.27	3.76 ± 1.18	0.65 ± 0.75	<0.001	0.182
	Δ at 3mo	−1.09 ± 1.65 *	−2.31 ± 1.51 *	−1.66 ± 1.69 *			0.004
OABSS	Baseline	4.52 ± 3.52	5.32 ± 3.20	4.85 ± 3.40	0.65 ± 0.67	<0.001	0.290
	Δ at 3mo	−2.06 ± 3.77 *	−1.69 ± 3.63 *	−1.89 ± 3.68 *			0.696
Uroflowmetry							
BC (mL)	Baseline	301.15 ± 179.27	263.83 ± 119.17	286.59 ± 158.78	441.71 ± 156.22	<0.001	0.302
	Δ at 3mo	25.55 ± 173.73	−14.17 ± 85.88	6.4 ± 138.76			0.280
Qmax (mL/s)	Baseline	9.72 ± 3.43	8.05 ± 4.14	9.07 ± 3.79	24.72 ± 7.33	<0.001	0.051
	Δ at 3mo	1.09 ± 5.41	6.06 ± 5.68 *	3.49 ± 6.04 *			0.001
cQmax	Baseline	0.6 ± 0.21	0.53 ± 0.28	0.57 ± 0.24	1.30 ± 0.72	<0.001	0.225
	Δ at 3mo	0.03 ± 0.28	0.4 ± 0.37 *	0.21 ± 0.37 *			<0.001
Vol (mL)	Baseline	259.86 ± 131.37	213.81 ± 97.49	241.89 ± 120.78	425.1 ± 153.52	<0.001	0.092
	Δ at 3mo	13.1 ± 116.06	14.7 ± 80.43	13.88 ± 99.58			0.953
PVR (mL)	Baseline	41.3 ± 105.39	50.03 ± 54.45	44.71 ± 88.72	16.65 ± 21.62	0.342	0.667
	Δ at 3mo	12.45 ± 198.92	−28.85 ± 60.24	−7.46 ± 149.31			0.305
VE	Baseline	0.89 ± 0.11	0.83 ± 0.16	0.87 ± 0.13	0.95 ± 0.08	0.005	0.086
	Δ at 3mo	−0.03 ± 0.21	0.08 ± 0.18 *	0.02 ± 0.2			0.054

*: *p* < 0.05 when compared with baseline data. $: *p*-value between overall clinical BPH patients and controls. #: *p*-value between medical treatment group, and surgical treatment group of clinical BPH patients. Δ: change in [specific variable]. BPH, benign prostatic hyperplasia; PSA, prostate-specific antigen; TPV, total prostate volume; TZI, transitional zone index; DWT, detrusor wall thickness; BWT, bladder wall thickness; IPSS, International Prostate Symptom Score; IPSS-S, IPSS storage subscore; IPSS-V, IPSS voiding subscore; QoL, quality of life; OABSS, overactive bladder symptoms score; BC, bladder capacity; Qmax, maximal urinary flow rate; cQmax, corrected maximal urinary flow rate; Vol, voided volume; PVR, post-void residual urine; and VE, voiding efficiency.

**Table 2 ijms-26-06516-t002:** Urinary biomarker levels between clinical BPH patients and controls.

			Clinical BPH				
		Medical Treatment Group N = 33	Surgical Treatment Group N = 29	Overall N = 62	Control N = 20	*p*-Value $	*p*-Value #
Urine biomarkers @							
8-OHdG	Baseline	79.83 (29.65, 110.22)	102.65 (39.43, 132.08)	85.31 (36.51, 118.61)	65.1 (54.95, 89.19)	0.136	0.505
	Δ at 3mo	−1.05 (−41.38, 13.44)	9.45 (−26.23, 47.85)	1.75 (−40.18, 31.69)			0.121
8-isoprostane	Baseline	11.88 (7.59, 26.73)	16.6 (9.56, 30.2)	14.16 (8.98, 29.82)	20.72 (12.86, 30.68)	0.057	0.266
	Δ at 3mo	−2.88 (−8.18, 2.73)	−4.12 (−21.24, 12.31)	−4.11 (−14.03, 4.66) *			1.000
TAC	Baseline	262.65 (179.62, 446.77)	316.41 (198.45, 585.2)	282.56 (195, 489.78)	167.03 (103.82, 416.68)	0.045	0.266
	Δ at 3mo	−23.18 (−136.51, 65.94)	−64.7 (−443.65, 65.08) *	−47.81 (−207.97, 64.13) *			0.605
PGE2	Baseline	166.95 (119.72, 301.06)	271.32 (185.2, 409.52)	218.92 (124.77, 339.98)	128.58 (87.85, 367.99)	0.003	0.004
	Δ at 3mo	−29.6 (−110.92, 75.02)	−64.71 (−223.72, 49.56) *	−39.84 (−157.77, 57.78)			0.605
IL-1β	Baseline	0.55 (0.41, 0.61)	0.48 (0.36, 0.55)	0.54 (0.39, 0.58)	0.5 (0.44, 0.51)	0.002	0.009
	Δ at 3mo	−0.06 (−0.14, 0.02) *	−0.05 (−0.1, 0.06)	−0.06 (−0.11, 0.04) *			1.000
IL-6	Baseline	0.64 (0.35, 1.88)	1.17 (0.37, 3.39)	0.73 (0.35, 2.36)	0.3 (0.16, 0.73)	0.045	0.266
	Δ at 3mo	−0.09 (−0.57, 0.1)	0.4 (−1.24, 6.73)	−0.04 (−0.61, 1.79)			0.121
IL-8	Baseline	1.28 (0.48, 3.15)	1.31 (0.28, 5.56)	1.28 (0.36, 3.63)	0.27 (0.14, 1.12)	0.136	0.824
	Δ at 3mo	−0.37 (−1.65, 0.38) *	1.07 (−0.85, 15.94)	−0.05 (−1.34, 1.54)			0.121
TNF-α	Baseline	1.08 (0.95, 1.25)	1.06 (0.31, 1.22)	1.08 (0.69, 1.24)	1.17 (1.11, 1.17)	0.234	0.824
	Δ at 3mo	−0.1 (−0.22, 0.04) *	0 (−0.16, 0.13)	−0.05 (−0.16, 0.09)			0.121

*: *p* < 0.05 when compared with baseline data. $: *p*-value between overall clinical BPH patients and controls. #: *p*-value between medical treatment group, and surgical treatment group of clinical BPH patients. Δ: change in [specific variable]. @: units are all pg/mL, except ng/mL in 8-OHdG, and nmol/μL in TAC 8-OHdG, 8-hydroxy-2-deoxyguanosine. TAC, total antioxidant capacity; PGE2, prostaglandin E2; and TNF-α, tumor necrosis factor-α.

**Table 3 ijms-26-06516-t003:** Clinical characteristics, symptom scores, uroflowmetry data, and urinary biomarker levels in the medical treatment group of clinical BPH patients with different treatment outcomes.

		Medical Treatment Group of Clinical BPH Patients			
		(A) GRA < 2 (N = 12, 36.4%)	(B) GRA ≥ 2 (N = 21, 63.6%)	Overall N = 33	*p*-Value
Age		65.58 ± 7.65	66 ± 6.46	65.85 ± 6.8	0.869
PSA (ng/mL)		3.79 ± 2.78	2.27 ± 1.68	2.82 ± 2.23	0.059
TPV (mL)		47.51 ± 18.42	42.19 ± 11.28	44.12 ± 14.25	0.310
TZI		0.43 ± 0.12	0.43 ± 0.13	0.43 ± 0.12	0.944
DWT (mm)		1.04 ± 0.5	1.38 ± 0.61	1.26 ± 0.59	0.117
BWT (mm)		2.46 ± 0.81	2.76 ± 0.79	2.65 ± 0.8	0.300
Symptom scores					
IPSS-V	Baseline	9.08 ± 5.14	12.05 ± 5.55	10.97 ± 5.51	0.140
	Δ at 3mo	−0.75 ± 4.03	−5.05 ± 6.39 *	−3.48 ± 5.96 *	0.044
IPSS-S	Baseline	6.08 ± 3.5	6.48 ± 3.97	6.33 ± 3.76	0.778
	Δ at 3mo	−0.25 ± 3.17	−2.86 ± 3.55 *	−1.91 ± 3.6 *	0.043
IPSS	Baseline	15.17 ± 7.53	18.52 ± 7.15	17.3 ± 7.35	0.212
	Δ at 3mo	−1 ± 4.82	−7.9 ± 7.14 *	−5.39 ± 7.16 *	0.006
QoL	Baseline	3.92 ± 1.31	3.67 ± 1.15	3.76 ± 1.2	0.573
	Δ at 3mo	−0.33 ± 1.5	−1.52 ± 1.6 *	−1.09 ± 1.65 *	0.044
OABSS	Baseline	5.75 ± 4.14	5.1 ± 3.39	5.33 ± 3.63	0.626
	Δ at 3mo	−0.58 ± 3.32	−2.9 ± 3.83 *	−2.06 ± 3.77 *	0.089
Uroflowmetry					
BC (mL)	Baseline	250.3 ± 126.14	286.19 ± 199.48	273.14 ± 175.06	0.579
	Δ at 3mo	26.22 ± 214.13	25.25 ± 158.66	25.55 ± 173.73	0.989
Qmax (mL/s)	Baseline	8.68 ± 3.36	9.6 ± 4.19	9.26 ± 3.88	0.520
	Δ at 3mo	−1.47 ± 2.9	2.24 ± 5.92	1.09 ± 5.41	0.088
cQmax	Baseline	0.58 ± 0.19	0.6 ± 0.25	0.59 ± 0.23	0.723
	Δ at 3mo	−0.04 ± 0.25	0.06 ± 0.29	0.03 ± 0.28	0.402
Vol (mL)	Baseline	229.33 ± 119.81	224.05 ± 86.25	225.97 ± 97.93	0.884
	Δ at 3mo	−64.78 ± 89.44	48.15 ± 110.98	13.1 ± 116.06	0.012
PVR (mL)	Baseline	21 ± 15.7	62.14 ± 158.54	47.18 ± 127.27	0.380
	Δ at 3mo	91 ± 238.18	−22.9 ± 173.74	12.45 ± 198.92	0.157
VE	Baseline	0.91 ± 0.08	0.85 ± 0.15	0.87 ± 0.13	0.205
	Δ at 3mo	−0.11 ± 0.21	0.01 ± 0.2	−0.03 ± 0.21	0.147
Urine biomarkers @					
8-OHdG	Baseline	66.09 (25.61, 93.36)	80.46 (29.1, 107.14)	79.83 (29.65, 110.22)	0.818
	Δ at 3mo	31.61 (0.3, 79.45)	−3.93 (−46.81, 3.89)	−1.05 (−41.38, 13.44)	0.017
8-isoprostane	Baseline	11.14 (3.37, 21.78)	11.34 (8.17, 20.74)	11.88 (7.59, 26.73)	0.818
	Δ at 3mo	−0.78 (−6.31, 12.72)	−4.63 (−13.31, −0.16) *	−2.88 (−8.18, 2.73)	0.376
TAC	Baseline	219.84 (151.79, 408.17)	249.88 (197.86, 457.04)	262.65 (179.62, 446.77)	0.818
	Δ at 3mo	32.71 (−99.69, 315.95)	−52.84 (−179.49, 22.5)	−23.18 (−136.51, 65.94)	0.378
PGE2	Baseline	147.56 (105.38, 298.02)	187.41 (133.92, 396.23)	166.95 (119.72, 301.06)	0.340
	Δ at 3mo	75.02 (−32.71, 338.89)	−60.04 (−128.32, 1.19) *	−29.6 (−110.92, 75.02)	0.102
IL-1β	Baseline	0.59 (0.42, 0.63)	0.55 (0.48, 0.64)	0.57 (0.45, 0.62)	0.611
	Δ at 3mo	−0.09 (−0.11, 0.02)	−0.06 (−0.17, 0.03) *	−0.06 (−0.14, 0.02) *	0.894
IL-6	Baseline	0.7 (0.23, 3.82)	0.69 (0.37, 2.14)	0.69 (0.37, 2.02)	0.795
	Δ at 3mo	−0.07 (−1.39, 1.44)	−0.15 (−0.57, 0.05)	−0.09 (−0.57, 0.1)	0.894
IL-8	Baseline	1.73 (0.29, 4.07)	1.22 (0.64, 9.47)	1.43 (0.55, 3.53)	0.611
	Δ at 3mo	−0.11 (−1.65, 0.67)	−0.39 (−2.91, 0.29) *	−0.37 (−1.65, 0.38) *	0.894
TNF-α	Baseline	1.19 (0.65, 1.3)	1.06 (0.96, 1.31)	1.16 (1, 1.29)	0.990
	Δ at 3mo	−0.11 (−0.22, 0.04) *	−0.06 (−0.36, 0.03)	−0.1 (−0.22, 0.04) *	0.537

*: *p* < 0.05 when compared with baseline data. Δ: change in [specific variable]. @: units are all pg/mL, except ng/mL in 8-OHdG, and nmol/μL in TAC. GRA, global response assessment; BPH, benign prostatic hyperplasia; PSA, prostate-specific antigen; TPV, total prostate volume; TZI, transitional zone index; DWT, detrusor wall thickness; BWT, bladder wall thickness; IPSS, International Prostate Symptom Score; IPSS-S, IPSS storage subscore; IPSS-V, IPSS voiding subscore; QoL, quality of life; OABSS, overactive bladder symptoms score; BC, bladder capacity; Qmax, maximal urinary flow rate; cQmax, corrected maximal urinary flow rate; Vol, voided volume; PVR, post-void residual urine; VE, voiding efficiency; 8-OHdG, 8-hydroxy-2-deoxyguanosine; TAC, total antioxidant capacity; PGE2, prostaglandin E2; and TNF-α, tumor necrosis factor-α.

## Data Availability

The original contributions presented in this study are included in the article/Appendix A. Further inquiries can be directed to the corresponding author.

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
