# Peer review of "Urinary Inflammatory and Oxidative Stress Biomarkers as Indicators for the Clinical Management of Benign Prostatic Hyperplasia"

_ijms, 2025, doi:10.3390/ijms26136516_

Round 1
Reviewer 1 Report
Comments and Suggestions for Authors
I added my comments in the revised manuscript attached

THE QUALITY IS MODEST
Author Response
Reviewer 1
- To my point of view, considering that the authors provides the evidence of the diagnostic and prognostic values of urinary inflammatory and oxidative stress biomarkers in managing clinical BPH patients, the authors should add to introduction section a brief overview of the importance of inflammatory biomarkers identified in bladder cancer patients by the invetsigation of specific bacteria and metabolites involved in the inflammatory and oxidative stress. I sugegst to read and cite the following paper Russo F. et al. Microorganisms 2024. 10.3390/microorganisms12102049
Reply: Thank you for your thoughtful suggestion to cite Russo et al. (Microorganisms, 2024). While we recognize that the study provides valuable insights into inflammation and oxidative stress in the context of bladder cancer, its focus differs substantially from the objectives and scope of our work. As our study specifically investigates urinary biomarkers in BPH, we feel that the reference is not directly relevant to our research. Therefore, we have respectfully decided not to include it in the current manuscript. We sincerely appreciate your recommendation.
- The author should cite the following paper that investigate the role of 8-OHdG as biomarkers to assess Oxidative Stress Damage Following Robot-Assisted Radical Prostatectomy (RARP). Di Minno et al.2022 10.3390/jcm11206102
Reply: Thank you for your valuable suggestion to cite Di Minno et al. (J Clin Med. 2022 Oct 17;11(20):6102). We agree that the study provides relevant insights into the use of 8-OHdG as a biomarker of oxidative stress. We have incorporated this reference into the revised manuscript accordingly. (ref. 20) (Line 230, Line 413-415)

Reviewer 2 Report
Comments and Suggestions for Authors
See the attached.

Author Response
Reviewer 2
Comments to the authors
Urinary Inflammatory and Oxidative Stress Biomarkers as Indicators for the Clinical Management of Benign Prostatic Hyperplasia
Title: Did the authors investigate how urinary inflammatory and oxidative stress biomarkers contribute to improved prognosis and their potential role in guiding clinical management?
Reply: Thank you for your comment. In our study, we investigated the associations between urinary inflammatory and oxidative stress biomarkers and clinical outcomes following both medical and surgical treatment for BPH. Our findings demonstrated that reductions in specific urinary biomarkers were significantly correlated with improvements in symptom scores and uroflowmetry parameters. These results suggest that such biomarkers may serve as potential indicators of disease severity and treatment response. While the primary aim was not to evaluate long-term prognosis or develop a clinical management algorithm, our data provide preliminary evidence supporting the potential utility of these biomarkers in guiding future individualized treatment strategies.
Abstract
Authors should comment on why they think there was an increase in the TAC levels in the BPH group?
The conclusion in the abstract is not justified by the results due to the following:
- No mention was made to disease severity in relation to pro-inflammatory cytokines and TAC concentration.
- In this case, TAC alone is not enough to justify the development of OS. Measuring prevalent reactive oxygen species such as H2O2, SO2, etc in this case will add more meaning to the result.
Alternatively, a ROS kit which allows for cumulative ROS measurement can be used to evaluate total. This will justify the findings on OS.
Reply: Thank you for your comments.
- Due to space constraints in the abstract, we focused on reporting the key findings, while the underlying pathophysiological mechanisms have been elaborated in the Discussion section.
- The associations between urinary biomarkers and clinical parameters were used to infer disease severity. Specifically, voiding efficiency (VE), a clinical indicator of voiding function, was negatively correlated with IL-1β, IL-6, and IL-8 levels prior to treatment. After treatment, changes in VE were negatively correlated with changes in IL-1β, and changes in postvoid residual urine were positively correlated with changes in IL-1β, IL-6, IL-8, and TNF-α. These correlations support the interpretation that higher levels of inflammatory biomarkers are associated with more severe symptoms and impaired voiding, thus reflecting disease severity.
- We fully acknowledge that TAC alone does not comprehensively capture the complexity of oxidative stress. However, in this study, we used several oxidative stress markers—8-OHdG, 8-isoprostane, and TAC—to represent different aspects of oxidative injury and antioxidant defense. Among these, only TAC showed significant increase in BPH groups, and significantly decreased after treatment. We have also added a statement in the limitations section acknowledging the need for additional assays in future studies to better delineate oxidative stress mechanisms.
“In addition, future studies should consider incorporating additional assays—such as direct quantification of ROS or the use of cumulative ROS detection kits—to better delineate oxidative stress mechanisms.” (Line 276-278)
Introduction
Lines 37-40 described the mechanisms of how BOO is driven by OS and inflammation. This concept should be explicitly elaborated to give the reader a preamble of what is currently known. Something along the below lines can be rephrased and cited accordingly.
Bladder Outlet Obstruction (BOO) is scientifically associated with hypoxia -related inflammation through a mechanism involving cyclic ischemia -reperfusion injury. Here's a more detailed explanation of the process:
- Increased Intravesical Pressure and Ischemia
BOO increases resistance to urine flow, leading to elevated intravesical pressure during voiding.
This increased pressure compresses intramural blood vessels, reducing bladder wall perfusion during bladder contraction.
As a result, ischemia (oxygen deprivation) occurs, especially in the detrusor muscle.
- Reperfusion After Voiding
Once voiding ends and bladder pressure drops, reperfusion (restoration of blood flow) occurs.
This leads to a cyclic pattern of ischemia and reperfusion, especially during repeated filling and voiding cycles.
- Oxidative Stress and Inflammation
The reperfusion phase generates reactive oxygen species (ROS) and free radicals, which damage cellular components.
ROS act as pro-inflammatory signals, activating inflammatory pathways, including:
NF-κB signaling
Upregulation of cytokines (e.g., TNF-α, IL-1β)
Recruitment of immune cells (macrophages, neutrophils)
- Tissue Remodeling and Fibrosis
Chronic exposure to hypoxia-reperfusion cycles causes persistent inflammation.
This stimulates fibroblast activation, collagen deposition, and smooth muscle hypertrophy.
The result is bladder wall remodeling, reduced compliance, and potentially, progression to upper urinary tract dysfunction.
Summary
In BOO, the cyclic ischemia-reperfusion injury due to elevated intravesical pressure leads to hypoxia, followed by oxidative stress and chronic inflammation. This inflammatory microenvironment promotes fibrosis and structural remodeling of the bladder tissue, contributing to long-term dysfunction.
Reply: Thank you for your insightful suggestion. We agree that the mechanistic pathway of cyclic ischemia-reperfusion injury in BOO deserves clear elaboration early in the manuscript. While these concepts are discussed across several sections in the original manuscript, we agree that briefly summarizing these mechanisms earlier in the manuscript will help readers better understand the context. Accordingly, we have revised the sentence to incorporate the key elements of cyclic ischemia-reperfusion injury, oxidative stress, and inflammation leading to tissue remodeling, as suggested.
Revised sentence (Line 38-44):
The development and progression of BOO and its associated urinary dysfunctions are profoundly influenced by cyclic ischemia-reperfusion injury, in which elevated intravesical pressure during voiding induces bladder wall ischemia, followed by reperfusion that generates oxidative stress and triggers hypoxia-related inflammation. This inflammatory microenvironment promotes fibrosis and structural remodeling of bladder tissue, contributing to long-term dysfunction.
Results
When describing the baseline clinical characteristics of the patients, it is important to mention both the BPH and control groups, and not only highlight the BPH group.
This helps the reader to have an idea of both groups without having to go to the table
Reply: Thank you for your comment. We have revised the text in the Results section to include a brief description of the baseline characteristics for both the BPH and control groups, allowing readers to grasp the comparison more easily without referring solely to the table.
Revised sentences (Line 84-93):
The study enrolled 62 clinical BPH patients (mean age 65.33±6.84 years) and 20 non-age-matched controls (mean age 38.0±7.9 years). Compared with controls, BPH patients exhibited significantly thicker bladder walls (bladder wall thickness [BWT]: 2.48 ±â€¯0.72 mm vs. 1.58 ±â€¯0.33 mm; detrusor wall thickness [DWT]: 1.13 ±â€¯0.48 mm vs. 0.69 ±â€¯0.20 mm), higher International Prostate Symptom Scores (IPSS: 17.42 ±â€¯7.11 vs. 1.20 ±â€¯1.20), lower maximal urinary flow rates (Qmax: 9.07 ±â€¯3.79 mL/sec vs. 24.72 ±â€¯7.33 mL/sec), lower corrected Qmax (cQmax: 0.57±0.24 vs. 1.30±0.72), smaller voided volumes (Vol: 241.89±120.78 mL vs. 425.1±153.52 mL), and reduced voiding efficiency (VE: 0.87±0.13 vs. 0.95±0.08). Prostate volume was measured only in BPH patients, with a mean of 42.76 ±â€¯14.5 mL (Table 1)
Lines 84-85: How about when compared to the control group?
Reply: Thank you for your thoughtful comment. As noted in Lines 84-93, the overall comparisons between BPH patients and controls have already been described in the Results section. In our study, we performed comparisons between the control group and the overall BPH group, as well as between the medical and surgical treatment subgroups. We chose not to perform direct comparisons between the control group and each subgroup individually to avoid unnecessary multiple testing, which could increase the risk of type I error and reduce statistical power.
To properly understand the role of urinary inflammatory and oxidative stress biomarkers in clinical BPH patients, it is important to state what is going on in the control group, and this should be done by comparing both groups.
Control Vs. Overall BPH
Control Vs. Medical cohort
Control Vs. surgical cohort
Medical Vs. surgical
Reply: Thank you for your thoughtful comment. As noted in Lines 109-111, we have described comparisons between the overall BPH group and the control group in the Results section. In this study, we focused on comparing the overall BPH group to controls, and the medical group to the surgical group (Line 113-114). We intentionally did not perform direct comparisons between each treatment subgroup and the control group to avoid unnecessary multiple testing, which could increase the risk of type I error and reduce statistical power.
Accordingly, the description of baseline urinary biomarker comparisons has been revised as follows:
“At baseline, urinary biomarker analysis showed that clinical BPH patients had significantly higher levels of TAC, prostaglandin E2 (PGE2), IL-1β, and IL-6 compared to controls (Table 2).” (Lines 109-111)
“Among clinical BPH patients, the surgical treatment group exhibited significantly higher baseline urinary PGE2 levels and lower IL-1β levels than the medical treatment group.” (Lines 113-114)
It is understandable that recruiting control patients within the age range of the case group might be challenging, however, the variation between the groups is too wide to ascertain justifiable comparison between the groups. Most especially because BPH is commonly reported in >55years.
To reduce the bias in result interpretation, was any statistical approach/strategy employed? If yes, kindly state what was done to reduce bias and misinterpretation.
Reply: We appreciate the reviewer’s thoughtful comment. We acknowledge that the age difference between the BPH and control groups is a limitation of our study. However, recruiting age-matched men with “truly normal” voiding function—defined by both low symptom scores and normal uroflowmetry—is particularly challenging, as LUTS prevalence increases with age. Therefore, our control group was selected based on strict functional criteria to best approximate normal lower urinary tract function, despite the younger age range. We will address the potential influence of age as a limitation in the revised manuscript.
“Third, our controls were stringently selected based on symptom scores and uroflowmetry to best approximate normal lower urinary tract function. As a result, age-matching was not performed, and the age difference between groups may introduce bias.” (Line 268-271)
In Table 1, was PSA, TPV and TZI not measured in the control group? If yes, authors should justify the rationale.
Reply: Thank you for your comment. PSA, TPV and TZI were not measured in the control group, as these participants were asymptomatic healthy individuals recruited for minor urological procedures without clinical indications for prostate evaluation. To avoid unnecessary procedures and ethical concerns in this population, we limited invasive or additional diagnostic assessments to those clinically justified.
A major concern in Table 2 is that there was no follow up measurement for the control group at 3 months to ascertain that nothing has changed within the control group.
Reply: Thank you for your comment. As the control group comprised asymptomatic individuals without lower urinary tract symptoms and no planned clinical intervention, follow-up sampling at 3 months was not performed to avoid unnecessary testing and participant burden. This design aimed to minimize unnecessary procedures and reduce participant burden, but limits interpretation of intra-individual biomarker variability over time in controls. This point has been added to the Limitation section.
“Fourth, follow-up urinary biomarker measurements were not performed in the control group, as these participants were asymptomatic and not undergoing any clinical intervention. This design aimed to minimize unnecessary procedures and reduce participant burden, but limits interpretation of intra-individual biomarker variability over time in controls.” (Line 271-275)
Discussion
The authors have appropriately highlighted and compared their findings with those of previous studies. However, the discussion section would benefit from a deeper exploration of the unique contributions this study makes to the existing literature. Specifically, the authors should provide a brief but clear explanation of the underlying mechanisms by which their findings align with or support earlier research. Additionally, it is important to articulate how these results offer novel insights or stand out from prior work. This would help to better position the study within the broader academic context and underscore its significance.
Reply: Thank you for your comment. As recommended, we have revised the paragraph of the Discussion to better highlight the unique contributions of our study in the context of existing literature.
The revised paragraph now reads as follows:
This study supports the hypothesis of the impact of BOO on bladder wall remodeling, delineated into three distinct phases: hypertrophy, compensation, and decompensation [5]. Tissue hypoxia, driven by cyclic ischemia-reperfusion injury, plays a critical role in this progression by activating inflammatory pathways and influencing signaling pathways related to angiogenesis, cell proliferation, and extracellular matrix remodeling. These molecular and physiological alterations result in morphological changes and functional impairments in bladder voiding, which may be reflected by changes in urinary biomarker profiles. (Line 250-257)
“Our findings align with this pathophysiological model, as elevated urinary levels of inflammatory markers in BPH patients were associated with impaired voiding function, such as lower VE. Notably, distinct urinary biomarker response patterns were observed between medical treatment and surgical treatment groups, suggesting different mechanisms of therapeutic effect—primarily inflammation resolution following medical therapy and oxidative stress reduction after surgical de-obstruction.” (Line 257-262)

Reviewer 3 Report
Comments and Suggestions for Authors
This study investigates the roles of urinary inflammatory and oxidative stress biomarkers in benign prostatic hyperplasia (BPH) patients, based on the premise that oxidative stress and hypoxia-induced inflammation contribute to BPH progression. The authors demonstrate that urinary inflammatory and oxidative stress biomarkers may serve as non-invasive indicators of disease severity and treatment response in clinical BPH management. However, several issues require attention to strengthen the manuscript.
Major Comments:
- Study design clarity: An enrollment flowchart is essential to illustrate patient selection, exclusion criteria, and group allocation processes.
- Sample size justification: A power analysis calculation determining the required sample size for each group should be included in the Methods section to validate the statistical power of the study.
- Age matching concern: The control group enrollment criteria (age ≥20 years) creates a significant age discrepancy with the BPH group, potentially confounding the results. Age-matched controls would strengthen the study design.
- Biomarker distribution analysis: In Figures 1A-C and 2A-B, the majority of IL-1β, IL-6, and IL-8 levels cluster near zero, which may compromise the validity of correlation analyses. The authors should address whether this skewed distribution pattern affects statistical interpretations and consider alternative analytical approaches, such as non-parametric tests or data transformation methods.
- Treatment discussion gap: In the Discussion section (Page 8, lines 198-202), the authors should explain why bladder detrusor thickness (BDT) remained unchanged following treatment, as this finding warrants mechanistic discussion.
Minor Comments:
- Terminology: Consider changing "Medical group" or "Medication group" to "Medicine group" for consistency with standard medical terminology.
- Statistical clarity: In Table 1, please specify what the P-value represents (e.g., comparison between groups, trend analysis).
- Uroflowmetry results: In Table 3, clarify whether there are statistically significant differences between baseline and post-treatment uroflowmetry parameters, as the significance markers are unclear.
- Treatment protocol specification: In the Methods section (Page 9, lines 273-274), while alpha blockers are mentioned as the 3-month treatment for the medical group, please clarify the role of 5-alpha reductase inhibitors in the treatment protocol.
Author Response
Reviewer 3
This study investigates the roles of urinary inflammatory and oxidative stress biomarkers in benign prostatic hyperplasia (BPH) patients, based on the premise that oxidative stress and hypoxia-induced inflammation contribute to BPH progression. The authors demonstrate that urinary inflammatory and oxidative stress biomarkers may serve as non-invasive indicators of disease severity and treatment response in clinical BPH management. However, several issues require attention to strengthen the manuscript.
Major Comments:
- Study design clarity: An enrollment flowchart is essential to illustrate patient selection, exclusion criteria, and group allocation processes.
Reply: Thank you for your helpful comment. As suggested, we have added an enrollment flowchart. This has been included as Supplementary Figure 4 in the revised manuscript. (Supplementary Figure 4.) (Line 459-461)
- Sample size justification: A power analysis calculation determining the required sample size for each group should be included in the Methods section to validate the statistical power of the study.
Reply: Thank you for your valuable suggestion. As recommended, we have added a post-hoc power analysis to the Methods section to assess the statistical power of our sample size.
“Post-hoc power calculation was performed in the biomarker with significant difference between the study and control groups” (Line 351- 352)
In the Results section, we now report:
“Post-hoc power analysis reported 8.7, 18, 65.1, and 48.8% power (with alpha value of 0.05) in the evaluation of TAC, PGE2, IL-1β, and IL-6 levels, respectively.” (Line 111-112)
- Age matching concern: The control group enrollment criteria (age ≥20 years) creates a significant age discrepancy with the BPH group, potentially confounding the results. Age-matched controls would strengthen the study design.
Reply: We appreciate the reviewer’s thoughtful comment. We acknowledge that the age difference between the BPH and control groups is a limitation of our study. However, recruiting age-matched men with “truly normal” voiding function—defined by both low symptom scores and normal uroflowmetry—is particularly challenging, as LUTS prevalence increases with age. Therefore, our control group was selected based on strict functional criteria to best approximate normal lower urinary tract function, despite the younger age range. We will address the potential influence of age as a limitation in the revised manuscript.
“Third, our controls were stringently selected based on symptom scores and uroflowmetry to best approximate normal lower urinary tract function. As a result, age-matching was not performed, and the age difference between groups may introduce bias.” (Line 268-271)
- Biomarker distribution analysis: In Figures 1A-C and 2A-B, the majority of IL-1β, IL-6, and IL-8 levels cluster near zero, which may compromise the validity of correlation analyses. The authors should address whether this skewed distribution pattern affects statistical interpretations and consider alternative analytical approaches, such as non-parametric tests or data transformation methods.
Reply: Thank you for your thoughtful comment. We acknowledge that IL-1β, IL-6, and IL-8 exhibited right-skewed, non-normal distributions. To address this, we conducted additional correlation analyses using Spearman’s rank correlation, a non-parametric method suitable for such distributions. While the Spearman’s results did not reach statistical significance, they showed similar directional trends to those of the original Pearson’s correlation.
Figure 1A-1C: (VE vs IL-1β, rho 0.155, p value 0.168), (VE vs IL-6, rho -0.139, p value 0.168), (VE vs IL-8, rho -0.208, p value 0.063
Figure 2A-2B: (changes in VE vs changes in IL-1β, rho -0.018, p value 0.894), (changes in PVR vs changes in IL-1β, rho 0.060, p value 0.659)
These findings suggest that although the strength of associations may be attenuated by data distribution, the observed relationships remain biologically relevant. We have revised the Methods and Results sections accordingly to clarify the statistical approach.
Revised Methods Section:
Correlation between clinical parameters and urinary biomarker levels was assessed using Pearson correlation analysis, and additionally confirmed using Spearman’s rank correlation for biomarkers exhibiting non-normal, right-skewed distributions (e.g., IL-1β, IL-6, and IL-8). (Line 348-351)
Revised Results Section:
Given the non-normal and right-skewed distribution of IL-1β, IL-6, and IL-8 observed in Figures 1A–1C and 2A–2B, additional analyses using Spearman’s rank correlation were performed. These supported the directionality of the original findings, although statistical significance was not reached, likely reflecting weaker associations under non-parametric analysis. (Line 160-164)
- Treatment discussion gap: In the Discussion section (Page 8, lines 198-202), the authors should explain why bladder detrusor thickness (BDT) remained unchanged following treatment, as this finding warrants mechanistic discussion.
Reply: Thank you for your comment. In our study, bladder wall thickness (BWT) and detrusor wall thickness (DWT) were measured only at baseline. Post-treatment measurements of these parameters were not performed, and thus, no conclusions can be drawn regarding changes following treatment. Furthermore, even if these measurements had been repeated, the relatively short follow-up period (3 months) may have been insufficient to capture measurable changes in bladder wall structure.
Minor Comments:
- Terminology: Consider changing "Medical group" or "Medication group" to "Medicine group" for consistency with standard medical terminology.
Reply: Thank you for your suggestion. To ensure consistency and clarity, we have standardized the terminology throughout the manuscript by using “medical treatment group,” which aligns with common usage in clinical research literature.
- Statistical clarity: In Table 1, please specify what the P-value represents (e.g., comparison between groups, trend analysis).
Reply: Thank you for your comment. In Table 1, we have clarified the meaning of each P-value as follows:
- The first P-value column (marked with a superscript “$”) represents statistical comparisons between the overall clinical BPH group and the control group.
- The second P-value column (marked with a superscript “#”) indicates comparisons between the medical treatment group and the surgical treatment group of clinical BPH patients.
We have also added clarifying footnotes below Table 1 to specify these comparisons
- Uroflowmetry results: In Table 3, clarify whether there are statistically significant differences between baseline and post-treatment uroflowmetry parameters, as the significance markers are unclear.
Reply: Thank you for pointing this out. In Table 3, we did not mark asterisks (*) for the baseline versus post-treatment uroflowmetry parameters within each treatment group because the within-group changes did not reach statistical significance (all p ≥ 0.05). This is likely due to the limited sample size after stratifying into medical and surgical treatment groups, which may reduce the statistical power to detect significant changes.
- Treatment protocol specification: In the Methods section (Page 9, lines 273-274), while alpha blockers are mentioned as the 3-month treatment for the medical group, please clarify the role of 5-alpha reductase inhibitors in the treatment protocol.
Reply: Thank you for your comment. In this study, patients in the medical treatment group received only alpha-blocker therapy (such as tamsulosin or silodosin) for 3 months. 5-alpha reductase inhibitors were not included in the treatment protocol. We have clarified this point in the revised Methods section for better understanding. (Line 305-306)

Round 2
Reviewer 3 Report
Comments and Suggestions for Authors
Thank you for the authors' detailed response and thorough revision of the manuscript. However, I still have some concerns regarding the statistical analysis methods section that require clarification:
1. Inconsistent statistical test selection: The authors state they used "analysis of variance" for clinical data comparisons. However, ANOVA is designed for comparing means across three or more groups, while this study appears to conduct exclusively two-group comparisons (study vs. control, medical treatment vs. surgical treatment groups). The authors should clarify whether independent t-tests were used for two-group comparisons or specify which analyses actually involved multiple groups requiring ANOVA.
2. Vague terminology: The phrase "nonparametric median tests" is too vague and does not specify the actual statistical procedure employed. The authors should explicitly name the tests used (e.g., Mann-Whitney U test for two-group comparisons, Kruskal-Wallis test for multiple groups) to ensure methodological transparency and reproducibility.
3. Unclear correlation analysis strategy: The correlation approach lacks methodological clarity. The authors mention using "Pearson correlation analysis, and additionally confirmed using Spearman's rank correlation" for certain biomarkers, which suggests a sequential rather than systematic approach. It would be more appropriate to state that normality was first assessed for all variables, and correlation methods were then selected based on data distribution characteristics, rather than using one method to "confirm" results from another.
Author Response
Reviewer:
Thank you for the authors' detailed response and thorough revision of the manuscript. However, I still have some concerns regarding the statistical analysis methods section that require clarification:
- Inconsistent statistical test selection: The authors state they used "analysis of variance" for clinical data comparisons. However, ANOVA is designed for comparing means across three or more groups, while this study appears to conduct exclusively two-group comparisons (study vs. control, medical treatment vs. surgical treatment groups). The authors should clarify whether independent t-tests were used for two-group comparisons or specify which analyses actually involved multiple groups requiring ANOVA.
Reply: Thank you for your insightful comment. We apologize for the oversight. You are correct that our study primarily involved comparisons between two groups (e.g., clinical BPH vs. controls, medical treatment vs. surgical treatment). We have revised the associated sentences:
“Continuous clinical data were compared using independent t-tests for all two-group comparisons (e.g., study vs. control, medical treatment vs. surgical treatment).” (Line 344- 346).
- Vague terminology: The phrase "nonparametric median tests" is too vague and does not specify the actual statistical procedure employed. The authors should explicitly name the tests used (e.g., Mann-Whitney U test for two-group comparisons, Kruskal-Wallis test for multiple groups) to ensure methodological transparency and reproducibility.
Reply: Thank you for your insightful comment. We agree that the original phrasing was too vague. We have now revised the Methods section to clearly state the statistical test used:
“The levels of urinary biomarkers for comparison were analyzed using the Mann–Whitney U test for two-group comparisons.” (Line 347)
- Unclear correlation analysis strategy: The correlation approach lacks methodological clarity. The authors mention using "Pearson correlation analysis, and additionally confirmed using Spearman's rank correlation" for certain biomarkers, which suggests a sequential rather than systematic approach. It would be more appropriate to state that normality was first assessed for all variables, and correlation methods were then selected based on data distribution characteristics, rather than using one method to "confirm" results from another.
Reply: Thank you for your helpful suggestion. We agree that the original description lacked clarity regarding our correlation analysis approach. In response, we have revised the text to reflect a more systematic and data-driven methodology:
“Prior to correlation analysis, the normality of variable distributions was assessed. Pearson correlation analysis was applied for normally distributed variables, while Spearman’s rank correlation was used for variables with non-normal, right-skewed distributions (e.g., IL-1β, IL-6, and IL-8).” (Line 347 – 351)
